# Arterial Calcifications in Patients with Liver Cirrhosis Are Linked to Hepatic Deficiency of Pyrophosphate Production Restored by Liver Transplantation

**DOI:** 10.3390/biomedicines10071496

**Published:** 2022-06-24

**Authors:** Audrey Laurain, Isabelle Rubera, Micheline Razzouk-Cadet, Stéphanie Bonnafous, Miguel Albuquerque, Valérie Paradis, Stéphanie Patouraux, Christophe Duranton, Olivier Lesaux, Georges Lefthériotis, Albert Tran, Rodolphe Anty, Philippe Gual, Antonio Iannelli, Guillaume Favre

**Affiliations:** 1Department of Nephrology, Pasteur 1 University Hospital, 06001 Nice, France; laurain.a@chu-nice.fr; 2Faculty of Medicine, Tour Pasteur, 28 Avenue de Valombrose, University of Côte d’Azur, 06000 Nice, France; isabelle.rubera@univ-cotedazur.fr (I.R.); stephanie.bonnafous@unice.fr (S.B.); patouraux.s@chu-nice.fr (S.P.); christophe.duranton@univ-cotedazur.fr (C.D.); leftheriotis.g@chu-nice.fr (G.L.); tran.a@chu-nice.fr (A.T.); anty.r@chu-nice.fr (R.A.); philippe.gual@unice.fr (P.G.); iannelli.a@chu-nice.fr (A.I.); 3LP2M CNRS UMR 7370, Tour Pasteur, 28 Avenue de Valombrose, 06000 Nice, France; 4Department of Nuclear Medicine, Archet 1 University Hospital, 06200 Nice, France; razzouk-cadet.m@chu-nice.fr; 5Team 8 “Chronic Liver Diseases Associated with Obesity and Alcohol” Inserm, U1065, Centre Méditerranéen de Médecine Moléculaire (C3M) Bâtiment Universitaire ARCHIMED? 151 Route Saint Antoine de Ginestière BP 2 3194, 06204 Nice, France; 6Digestive Unit, Archet 2 University Hospital, 06200 Nice, France; 7Pathology Department, Beaujon University Hospital, AP-HP, 92110 Clichy, France; miguel.albuquerque@aphp.fr (M.A.); valerie.paradis@aphp.fr (V.P.); 8Inserm U1149, Beaujon University Hospital, 92110 Clichy, France; 9Pathology Department, Pasteur 1 University Hospital, 06000 Nice, France; 10Department Cell and Molecular Biology, John A. Burns School of Medicine, University of Hawaii, Honolulu, HI 96813-5534, USA; lesaux@hawaii.edu; 11Department of Vascular Medicine and Surgery, Pasteur 1 University Hospital, 06000 Nice, France

**Keywords:** arterial calcification, pyrophosphate, liver fibrosis

## Abstract

Liver fibrosis is associated with arterial calcification (AC). Since the liver is a source of inorganic pyrophosphate (PPi), an anti-calcifying compound, we investigated the relationship between plasma PPi ([PPi]pl), liver fibrosis, liver function, AC, and the hepatic expression of genes regulating PPi homeostasis. To that aim, we compared [PPi]pl before liver transplantation (LT) and 3 months after LT. We also assessed the expression of four key regulators of PPi in liver tissues and established correlations between AC, and scores of liver fibrosis and liver failure in these patients. LT candidates with various liver diseases were included. AC scores were assessed in coronary arteries, abdominal aorta, and aortic valves. Liver fibrosis was evaluated on liver biopsies and from non-invasive tests (FIB-4 and APRI scores). Liver functions were assessed by measuring serum albumin, ALBI, MELD, and Pugh–Child scores. An enzymatic assay was used to dose [PPi]pl. A group of patients without liver alterations from a previous cohort provided a control group. Gene expression assays were performed with mRNA extracted from liver biopsies and compared between LT recipients and the control individuals. [PPi]pl negatively correlated with APRI (r = −0.57, *p* = 0.001, n = 29) and FIB-4 (r = −0.47, *p* = 0.006, n = 29) but not with interstitial fibrosis index from liver biopsies (r = 0.07, *p* = 0.40, n = 16). Serum albumin positively correlated with [PPi]pl (r = 0.71; *p* < 0.0001, n = 20). ALBI, MELD, and Pugh–Child scores correlated negatively with [PPi]pl (r = −0.60, *p* = 0.0005; r = −0.56, *p* = 0.002; r = −0.41, *p* = 0.02, respectively, with n = 20). Liver fibrosis assessed on liver biopsies by FIB-4 and by APRI positively correlated with coronary AC (r = 0.51, *p* = 0.02, n = 16; r = 0.58, *p* = 0.009, n = 20; r = 0.41, *p* = 0.04, n = 20, respectively) and with abdominal aorta AC (r = 0.50, *p* = 0.02, n = 16; r = 0.67, *p* = 0.002, n = 20; r = 0.61, *p* = 0.04, n = 20, respectively). FIB-4 also positively correlated with aortic valve calcification (r = 0.40, *p* = 0.046, n = 20). The key regulator genes of PPi production in liver were lower in patients undergoing liver transplantation as compared to controls. Three months after surgery, serum albumin levels were restored to physiological levels (40 [37–44] vs. 35 [30–40], *p* = 0.009) and [PPi]pl was normalized (1.40 [1.07–1.86] vs. 0.68 [0.53–0.80] µmol/L, *p* = 0.0005, n = 12). Liver failure and/or fibrosis correlated with AC in several arterial beds and were associated with low plasma PPi and dysregulation of key proteins involved in PPi homeostasis. Liver transplantation normalized these parameters.

## 1. Introduction

Arterial calcification (AC) is a strong cardiovascular risk factor in the general population [1] and more so in recipients of liver transplant [2]. Arterial calcification can be found in the intimal and/or in the medial layers of large arteries and also affects cardiac valves. Ectopic calcification in the vasculature is a dynamic process [3] and is usually assessed by computer tomogram (CT) scan as described by Agatston [4]. In the last decade, liver fibrosis has been associated with AC [5,6,7]. Liver fibrosis is characterized by the hepatic accumulation of extracellular matrix resulting from diverse chronic liver pathologies including viral hepatitis, non-alcoholic steatohepatitis (NASH), and other causes [8]. Liver cirrhosis is the most advanced stage of liver fibrosis and is histologically characterized by regenerative nodules of liver parenchyma surrounded by fibrotic septa. It can be complicated by portal hypertension and liver function deterioration with the related complications such as ascites, variceal bleeding, or hepatic encephalopathy [8]. Plasma alkaline phosphatase (ALP) activity results from a mixture of bone and liver ALP, which are produced by the same *ALPL* gene encoding tissue non-specific ALP (TNAP). In the absence of bone disease and vitamin D deficiency, a mild increase of plasma ALP activity is observed in many liver diseases, and a marked elevation of plasma ALP activity is the signpost of the primary cholestatic diseases [9,10]. Liver cirrhosis can be associated with high levels of total bilirubin and gamma-glutamyl transferase (GGT) [11] and eventually with liver failure, resulting in low prothrombin levels and/or hypoalbuminemia [12]. Liver cirrhosis with end stage liver failure and/or hepatocellular carcinoma may be treated with liver transplantation (LT).

The association between AC and liver fibrosis may be explained by the presence of similar factors. Inflammation, accumulation of pro-inflammatory macrophages and myofibroblasts, and increased oxidative stress are recognized as causal factors involved in the pathogenesis of these two related pathological conditions [3,8]. An imbalance between pro and anti-calcifying compounds likely plays an important role, as the liver is the main source of fetuin A [13] and osteoprotegerin [14], which are upregulated with liver dysfunction. Inorganic pyrophosphate (PPi) is one of the main regulators of soft tissues calcification and is found at the micromolar range in plasma [15,16,17]. The liver is responsible for the majority of the circulating PPi [15,18]. PPi prevents the formation of hydroxyapatite crystals that results from calcium and inorganic phosphates precipitates. Physiological mineralization is normally restricted to bones and teeth. However, in specific pathological contexts, ectopic calcification can affect a variety of soft tissues, but the skin, kidneys, tendons, and cardiovascular tissues are particularly prone to this pathology. Indeed, low levels of plasma PPi ([PPi]pl) drives AC in several monogenic diseases and most likely in the advanced stages of chronic kidney disease (CKD) [17,19]. In recent years, the identification of mutations in the ATP-binding cassette (ABC) transporter ABCC6 [20,21,22] and the characterization of its function [23] have provided new molecular insight into the regulation of ectopic calcification inhibition in relation to plasma PPi. [PPi]pl mainly depends on the liver expression of adenosine triphosphate (ATP)-binding cassette (ABC) subfamily C, member 6 (ABCC6) [15], which is primarily found in the liver [24]. ABCC6 promotes the cellular release of ATP from hepatocytes [15,23]. Extracellular ATP is then hydrolyzed into AMP and PPi by ectonucleotide pyrophosphatase/phosphodiesterase 1 (ENPP-1). ENPP-1 is a cell membrane glycoprotein, which is particularly abundant in hepatocytes and is the only enzyme generating PPi [25]. Ecto 5′ nucleotidase (NT5E as known as CD73) further hydrolyzes AMP into inorganic phosphates (Pi) and adenosine, which represses the transcription of *ALPL* gene encoding TNAP [26]. TNAP is a membrane-bound enzyme which is also present in a circulating form in the plasma. TNAP hydrolyzes PPi into Pi [27], and plasma ALP activity is a major determinant of systemic PPi [17]. It is therefore conceivable that liver dysfunction in humans may affect PPi homeostasis and, thus, promote the development of AC.

To test this hypothesis, we compared [PPi]pl before LT and 3 months after LT. We also assessed the expression of four key regulators of PPi in liver tissues and established correlations between AC, indices of liver fibrosis, and liver failure in these patients.

## 2. Materials and Methods

### 2.1. Patients

All prevalent and incident patients listed for LT were invited to take part in the study. Non-inclusion criteria were fulminant hepatitis or refusal to participate in the study. The study was conducted according to the guidelines of the Declaration of Helsinki and approved on 2 July 2018 and by the Ethics Committee: “Comité de Protection des Personnes Est-III”. Informed consents were obtained from all patients, and the study was registered in ClinicalTrials.gov accessed on 3 July 2018) with the following identifier: NCT03576859.

To constitute a group of liver biopsies without liver alterations, we took advantage of our cohort of morbidly obese patients. These patients were recruited through the Department of Digestive Surgery and Liver Transplantation (Nice, France) where they underwent bariatric surgery for their morbid obesity. All subjects gave informed written consent in accordance with French legislation on Ethics and Human Research referred to as Huriet–Serusclat’s law. The ‘‘Comité Consultatif de Protection des Personnes dans la Recherche Biomédicale de Nice” approved the study (07/04/2003, no. 03.017). Surgical liver biopsies were obtained during surgery, and no ischemic preconditioning was performed. Hepatic histopathological analysis was performed according to the scoring system of Kleiner et al. [28]. Liver samples were selected from patients without hepatitis B or hepatitis C infection, excessive alcohol consumption (>20 g/d), or another cause of chronic liver disease, as previously described [29,30,31]. Liver samples did not display any hepatic steatosis, inflammation, or fibrosis.

### 2.2. Trial Procedures

This was a pilot study following the standard course of medical and surgical care for LT candidates. Wait-listed patients were cared for according to the best clinical practices. A myocardial scintigraphy associated with a CT scan was used to measure AC. At the time of LT, liver biopsies from the explanted organ were obtained during surgery. Samples were immediately snap-frozen in liquid nitrogen before storage at −80 °C for later mRNA analyses or fixed paraformaldehyde to determine the histological fibrosis index. All LT recipients were examined three months post-surgery, when liver function is typically restored (Figure 1). Blood and urine samples were systematically collected in the morning after an overnight fast during the two scheduled visits, at baseline and three months after LT.

### 2.3. Calculation of Scores

Liver biopsy sections were stained with Masson’s trichrome, and images were collected with an AT Turbo (LEICA©, Wetzlar, Germany) scanscope. Liver fibrosis red–green–blue images were acquired using an ImageScope (LEICA©, Wetzlar, Germany) and analyzed using a dedicated algorithm. A ratio of green positive pixels to total pixels detected, representing the areas of liver fibrosis over total tissue section, was obtained and used as an index of interstitial fibrosis. FIB-4-index [32] was calculated using the following formula: [age (years) × AST (IU/L)]/[platelet counts (×10^9^/L) × (ALT (IU/L))^1/2^]. APRI (AST-to-platelet ratio) index [33] was calculated using the formula: [(AST/ULN)/platelet counts (x10^9^/L)] × 100 (ULN denotes the upper level of normal). The Pugh–Child score was determined as previously described [34]. The Albumin–Bilirubin (ALBI) score [35] was calculated with serum albumin and total bilirubin, and the Model for End-Stage Liver Disease (MELD) scores [36] were based on serum creatinine, total bilirubin, international normalized ratio, and serum sodium. AC scores were measured according to Agatston [4]. CT scans were performed with a Discovery NM/CT 670 with 16-slice scanner (Brightspeed Elite, Winter Park, FL, USA) integrating dose reduction technology (General Electric, Milwaukee, WI, USA) in a supine position following myocardial scintigraphy. The 4DM software (General Electric, Milwaukee, WI, USA) was used to calculate the AC scores. Briefly, the software selected 3 consecutive pixels on 2.5 mm-thick consecutive slices to determine points which density was measured in Hounsfield units (UHs). The points were positioned on the coronary arteries, abdominal aorta, and on the aortic valves using a hand-controlled trackball cursor. The points were multiplied by 1 (130 to 199 UH), 2 (200 to 299 UH), 3 (300 to 399 UH), or 4 (≥400 UH) and added to produce the AC scores. The AC scores of the abdominal aorta were measured between the diaphragm and the bifurcation of the iliac arteries and divided by the length of the aorta to allow for the comparison of subjects of different heights. AC scores were log-transformed for graphical representation and expressed in arbitrary units.

### 2.4. Specific Dosages

Plasma ALP activity was assessed by absorbance at 450 nm of paranitrophenol at alkaline pH. Inorganic pyrophosphate concentrations were measured using a modified method based on the enzyme assay described by Jansen et al. [15], as previously reported [17,37]. Briefly, PPi was converted into ATP using ATP sulfurylase (New England Biolabs, Ipswich, MA, USA, MO394L) in the presence of APS (adenosine-5′-phosphosulfate, Sigma-Aldrich, St. Louis, MO, USA, A5508). The generated ATP was subsequently quantified using a luminescent ATP detection kit (ATPlite, PerkinElmer, Waltham, MA, USA) according to the manufacturer’s instructions. The luminescence was measured on a microplate reader (Synergy HT, BioTek, Shoreline, WA, USA). PPi values were obtained by subtracting basal ATP levels measured in each sample.

Serum creatinine was assessed with Jaffe’s kinetic methods (IDMS standardized). Renal function was determined by estimating glomerular filtration rate according to the CKD–EPI formula with creatinine. Fractional excretion of PPi (FePPi) was calculated as a percentage according to the following formula: [PPi]_u_ × [creat]_pl_/[PPi]_pl_ × [creat]_u_ × 100. Body mass index (BMI) was calculated with the ratio of weight and square value of height.

For real-time quantitative RT-PCR analyses, total liver RNA was extracted using an RNeasy Mini Kit (74104, Qiagen, Hilden, Germany) and treated with Turbo DNA-free DNase (AM 1907, Thermo Fisher Scientific Inc., Waltham, MA, USA) following the manufacturer’s protocol. The quantity and quality of the RNA samples were determined using an Agilent 2100 Bioanalyzer with an RNA 6000 Nano Kit (5067–1511, Agilent Technologies, Santa Clara, CA, USA). Total RNA (1 µg) was reverse transcribed with a High-Capacity cDNA Reverse Transcription Kit (Thermo Fisher Scientific Inc.). Real-time quantitative PCR was performed in duplicate for each sample using a StepOne Plus Real-Time PCR System (Thermo Fisher Scientific Inc.), as previously described [29]. Gene expression was normalized to the RPLP0 (Ribosomal Phosphoprotein Large P0) housekeeping gene and calculated based on the comparative cycle threshold Ct method (2-ΔΔCt). The results were expressed relative to mRNA levels in “Patients without fibrosis”. Results were expressed as median ± interquartiles 25–75 and statistically analyzed using the Mann–Whitney test.

The sequences of the primers used with 2x SensiFAST SYBR HI-ROX mix (Bioline, London, UK) are presented in Table 1.

### 2.5. Outcomes

The primary outcome was a deficit in [PPi]pl before LT, defined by a statistically significant difference between [PPi]pl in LT candidates at baseline and [PPi]pl three months after LT.

The secondary outcomes were the assessments of:

1. Low [PPi]pl levels in LT candidates with the most severe liver fibrosis or with the most severe liver failure assessed by the following associations:Association between [PPi]pl and liver fibrosis indexes;Association between [PPi]pl and albumin or clinical scores of liver failure.

2. High arterial calcification levels in LT candidates with the most severe liver fibrosis or with the most severe liver failure assessed by the following associations:Association between [PPi]pl and AC indexes;Association between liver fibrosis indexes and AC indexes;Association between albumin or clinical scores of liver failure and AC indexes.

3. High levels of plasma ALP activity, responsible for [PPi]pl hydrolysis, and low levels of molecules involved in PPi production, assessed by:the correlation between plasma ALP activity and [PPi]pl at baseline;the comparison of mRNA levels of *ABCC6*, *ENPP1*, *ALPL*, and *NT5E* between the liver biopsies from the explanted liver from patients undergoing LT and liver biopsies from a control group of individuals without liver disease;the comparison between plasma ALP activity in LT candidates at baseline and plasma ALP activity in LT recipients three months after LT.

### 2.6. Statistical Analysis

#### 2.6.1. Sample Size Calculation

We hypothesized that [PPi]pl were low in LT candidates and were restored within the normal range three months after LT. We estimated, based on a previous study [17], that the participation of ten patients would sufficiently power the study to identify a significant difference in the median [PPi]pl before and after LT.

#### 2.6.2. Statistics

Quantitative data were presented as medians with interquartile ranges [25th–75th percentiles], and qualitative data were presented as counts and percentages. Data were compared using Mann–Whitney tests, paired or unpaired as appropriate. Correlations were evaluated with Spearman’s tests. All statistics were conducted with GraphPad Prism 6^®^ software, Ritme, Paris, France). A *p*-value < 0.05 was considered statistically significant.

## 3. Results

### 3.1. Description of the Study

Between November 2018 and November 2020, 29 LT candidates were included in the study (Figure 1). Baseline [PPi]pl and clinical liver fibrosis scores were determined in all 29 LT candidates. Sixteen patients underwent LT 8 months after initial consultation. During surgery, a biopsy of the explanted liver was taken and prepared for interstitial fibrosis scoring from all patients, and 10 samples were obtained for mRNA extraction. Of the 29 patients, 12 LT recipients reached the 3rd month visit post-surgery within the study period. A cohort of six individuals without significant liver disease was used as a control group for the gene expression experiments.

### 3.2. Description of the Patients

The median age of the 29 LT candidates was 60 [57–65] years. There were 22 men (76%) and 7 women (24%), and the median BMI was 27 [24–30] kg/m^2^ (Table 2). Recommendations for LT were based on liver cirrhosis with viral hepatitis in eight patients (27.5%), alcoholism in seven patients (24.0%), alcoholism and viral hepatitis in five patients (17.0%), non-alcoholic steatohepatitis (NASH) in two patients (7.0%), NASH and alcoholism in two patients (7%), cryptogenetic in two patients (7.0%), hemochromatosis in one patient (3.5%), and no liver cirrhosis with papillomatosis in one patient (3.5%) or refractory hepatocellular carcinoma in one patient (3.5%).

Twenty patients also had a hepatocellular carcinoma (69%), 21 patients were smokers (72%), and 8 had type 2 diabetes (T2D) (27%). The median interstitial fibrosis score was 12.2 [7.2–17.9], the median FIB-4 score was 4.7 [1.9–9.3], and the median APRI index was 1.3 [0.5–2.7]. The Pugh–Child score was A in 16 LT candidates (55%), B in 10 LT candidates (34%), and C in 3 LT candidates (11%). Liver function tests are given in Table 3.

The control group included six morbidly obese patients with no (*n* = 2) or mild steatosis (*n* = 4), without liver inflammation, and without septal fibrosis (Table 4). Median age was 26 [24–32] years. This group only included women (100%) with type 2 diabetes (100%), and their median BMI was 43 [40–46] kg/m^2^. Their median FIB-4 score was 0.43 [0.37–0.52], and their median APRI score was 0.23 [0.20–0.30].

1. Low [PPi]pl levels were associated with the most severe liver fibrosis before LT.

The APRI score correlated negatively with [PPi]pl (r = −0.57, *p* = 0.001), and the FIB-4 score correlated negatively with [PPi]pl (r = −0.47, *p* = 0.007) (Figure 2). However, there was no correlation between the interstitial fibrosis index and [PPi]pl (r = 0.07, *p* = 0.40, *n* = 16).

2. Low [PPi]pl levels were associated with the most severe liver failure before LT.

Serum albumin correlated positively with [PPi]pl (r = 0.71; *p* < 0.0001), and ALBI, MELD, and Pugh–Child scores correlated negatively with [PPi]pl (Figure 3).

3. LT completely restored plasma PPi levels.

The median [PPi]pl was significantly higher in LT recipients three months after LT than at baseline (1.40 [1.07–1.86] vs. 0.68 [0.53–0.80] µmol/L, *p* = 0.0005) (Table 1).

4. PPi homeostasis was impaired in hepatocytes before LT.

The expression of genes directly related to PPi production were significantly downregulated in the diseased livers of LT recipients as compared to controls: *ABCC6* (0.33 vs. 1.06, *p* = 0.01), *ENPP1* (0.19 vs. 0.89, *p* = 0.001). The mRNA levels of *ALPL* were unchanged (1.23 vs. 0.89, *p* = 0.11), whereas the expression of *NT5E* was significantly reduced (0.28 vs. 0.74, *p* = 0.0002) (Figure 4). We measured higher plasma ALP activity in LT candidates than in LT recipients (190 [87–321] vs. 81 [65–158] UI/L, *p* = 0.05) (Table 1). Plasma ALP activity negatively correlated with [PPi]pl at baseline (r = −0.54, *p* = 0.0003), as expected [17].

5. High AC scores were associated with the most severe liver fibrosis indexes in LTC.

The AC scores measured in the coronary arteries and in the abdominal aorta positively correlated with the interstitial fibrosis and with APRI and FIB-4 indices (Figure 5). The AC score of aortic valves also positively correlated with FIB-4 (r = 0.40, *p* < 0.05), but there was no correlation with the interstitial fibrosis index (r = 0.40, *p* = 0.064), or the APRI score (r = 0.30, *p* = 0.11). Not surprisingly, coronary AC (r = 0.41, *p* = 0.035) and abdominal aorta AC (r = 0.44, *p* = 0.03) correlated with age.

6. High AC scores correlated with the most severe liver failure in LTC.

Abdominal aorta AC scores positively correlated with MELD score (r = 0.46, *p* = 0.042) and with ALBI score (r = 0.53, *p* = 0.016). Aortic valve calcifications negatively correlated with albumin (r = −0.51, *p* = 0.021) and positively with the Pugh–Child score (r = 0.57, *p* = 0.008). AC, irrespective of its localization, and liver failure indices showed no significant correlation (data not shown). Finally, there were no correlations between [PPi]pl and coronary AC scores (r = −0.06, *p* = 0.40), or between [PPi]pl and abdominal aorta AC scores (r = −0.23, *p* = 0.17).

## 4. Discussion

Liver cirrhosis with liver failure of any grade is associated with a deficit in plasma PPi, which is the result of a significant reduction in the gene expression levels of key regulators of PPi synthesis in the liver, i.e., *ABCC6* and *ENPP1*, concomitant to enhanced degradation via TNAP. We found that LT restores [PPi]pl 3 months post-surgery to within physiological range [17,37]. This is likely due to the normalization of the expression levels of *ABCC6, ENPP1, NT5E*, and ALPL in the newly and healthy transplanted livers. The central role of the liver in regulating plasma PPi was demonstrated in animal studies. In liver perfusion experiments, Jansen et al. and Li et al. showed that this tissue produces the bulk of circulating PPi in mouse and rat, respectively [15,38]. Furthermore, plasma PPi levels are reduced in liver perfusates of *Abcc6^-/-^* mice as compared to wild-type controls, and the transient expression of the human ABCC6 protein in liver restores PPi concentration in liver perfusate to near normal levels [26]. Furthermore, the liver-specific inactivation of *Abcc6* reduces plasma PPi levels, which can be further decreased by the combined inactivation of *Enpp1* [18]. Our results with human LT patients are clearly consistent with these previously published data in animal models and confirm the intimate role played by the ABCC6➝ENPP1➝NT5E➝TNAP pathway in regulating plasma PPi homeostasis. Although we observed no significant variation in the expression of *ALPL* in diseased livers vs. controls in the limited number of samples we had access to, plasma ALP activity was abnormally elevated in LT candidates. ALP activity was significantly reduced 3 months post-surgery (Table 1), which is consistent with a restoration of adenosine production via NT5E in liver and other tissues (Figure 4B) [26,39]. Indeed, TNAP regulates ectopic calcification not only from liver but also from many other tissues [40,41,42].

Until now, ectopic calcification only in coronary arteries has been documented in patients with liver disease [5,6,7]. In this study, we not only confirmed the presence of AC in the coronary arteries of patients with liver fibrosis, but we found that calcification also affected the abdominal aorta and aortic valves (Figure 5). CT scans did not provide enough resolution to distinguish whether AC affected the medial or intimal layers of the arteries. However, the calcification of coronary arteries is usually found in the intima layers and is related to atherosclerosis and ischemic cardiopathy. In our cohort, we found that coronary AC was not associated with ischemic cardiopathy, therefore, it is safe to assume that calcification primarily affected the media. Similarly, in patients on maintenance hemodialysis (HD), severity of ischemic cardiopathy, as determined by coronary artery stenosis, is not associated with coronary AC scores [43]. In contrast, medial calcification is the hallmark of age and is significantly more pronounced in patients with TD2 or in patients on maintenance HD [44,45,46]. Although the leading pathomechanism is not fully known, an imbalance between pro-and anti-calcifying factors, especially PPi, is strongly suspected [46,47]. Unlike vascular mineralization, aortic valve calcification mainly occurs on degenerative valves and depends on mechanic factors such as arterial hypertension. In our study we found that the pattern of ectopic calcification in LT candidates mimics the observations in patients on maintenance HD [48,49]. These similarities argue for a key role of one or several circulating compound(s) in the formation of AC, probably with PPi as the main contributor [17,50,51].

Despite different causes of liver diseases in our patients, we observed a homogeneous calcification pattern, suggesting that liver fibrosis and/or liver failure associated with the dysregulation of genes involved in PPi homeostasis are the main drivers for AC. The role of liver fibrosis was pointed to in the literature, as shown in studies between cardiovascular mortality and non-alcoholic fatty liver disease (NAFLD) in the national health and nutrition examination survey III (NHANES III). In this representative population of the USA, the association between NAFLD and the occurrence of cardiovascular events with 14 years follow-up remained controversial [52,53], and NAFLD was not associated with an increased risk of cardiovascular death [52]. However, in this population, a high APRI index raised the hazard ratio for cardiovascular mortality to 2.53, and a high FIB-4 index elevated the hazard ratio to 2.68, compared to the patients with low fibrosis scores [54]. An association between AC and hepatic steatosis was found in larger cohorts of patients. In a much larger cohort than NHANES III, involving 10,308 patients with ultrasonographic detection of hepatic steatosis, the odds ratio for the presence of coronary AC was elevated in patients with alcoholic fatty liver disease (AFLD) or NAFLD as compared to patients without hepatic steatosis, with or without excessive alcoholic consumption [55]. Conversely, in patients with ischemic cardiopathy without liver fibrosis index assessment, the coronary AC scores increased with the prevalence of NAFLD [56].

In this work, we were able to show an association between liver fibrosis and AC in a small cohort of patients with a high fibrosis score (Figure 5). Similar associations between non-invasive fibrosis indexes and AC were described in larger cohorts of 81, 94, or 142 patients, respectively [5,6,7]. In one of these cohorts, patients had a lower FIB-4 index (2.09) [6] than our patients. Further, a large meta-analysis regarding the association of impaired liver function tests with AC among 1.23 million participants from diverse populations showed that one standard deviation in plasma ALP activity toward the higher values increased the risk of AC by up to 8%, whereas one standard deviation in GGT toward the higher values increased the risk of AC to 23% [11]. In light of these data, one may speculate a progressive effect of liver fibrosis and/or of liver function impairment on the development of AC.

The strength of our study is in the comparison of the plasma levels of PPi before and after LT, in the molecular analysis of genes involved in PPi homeostasis, in the measurement of fibrosis score on liver biopsies, and in the various localizations of AC (coronary and abdominal arteries, aortic valves). However, here are some limitations to our interpretations. We find no association between baseline [PPi]pl and interstitial fibrosis scores, whereas there is an association between baseline [PPi]pl and non-invasive fibrosis scores. This is probably due to the limited number of liver biopsies we had access to in order to determine interstitial fibrosis scores, as compared to those with blood tests for the measure of the non-invasive fibrosis scores (16 vs. 29 patients, respectively; Figure 1). We found no association between AC and [PPi]pl, most likely because the development of AC is a slow and progressive process spanning decades, whereas our study was limited to 3 months post-surgery. Therefore, longer longitudinal studies would be needed to determine the progression of AC in relation to [PPi]pl levels. However, one considers that the restoration of [PPi]pl does not lead to a decrease in existing calcification in animal models [24,57]. Furthermore, the small number of patients did not allow for adjustments for known confounding factors influencing AC, such as type 2 diabetes or age. In addition, control patients without liver disease could not be sex- and age-matched with LT recipients. Finally, we were not able to perform immunohistology analysis on liver samples.

In conclusion, we showed that liver diseases promote AC in several arterial beds via the dysregulation of key proteins involved in the production and degradation of plasma PPi, and that liver transplantation normalized these parameters.

## Figures and Tables

**Figure 1 biomedicines-10-01496-f001:**
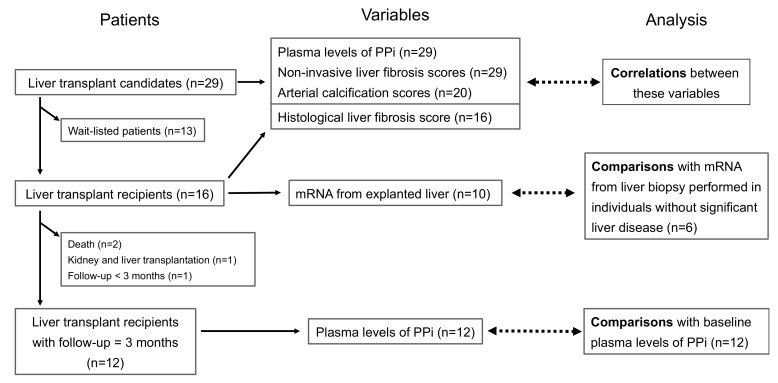
Flow chart and study plan. Legend: during the study period, 29 patients waiting for LT were included. At baseline, plasma PPi levels and liver fibrosis scores were determined in all patients; AC scores were determined in 20 LT candidates. Correlations between AC scores, fibrosis scores, and [PPi]pl were analyzed. Sixteen patients underwent LT. Biopsies from explanted liver prepared for interstitial fibrosis score were obtained from all patients, and 10 samples were prepared for mRNA extraction. Six patients without significant liver disease were used as controls for the gene expression assays. Twelve LT recipients reached the 3rd month post-surgery within the study period, and their [PPi]pl were compared with baseline values.

**Figure 2 biomedicines-10-01496-f002:**
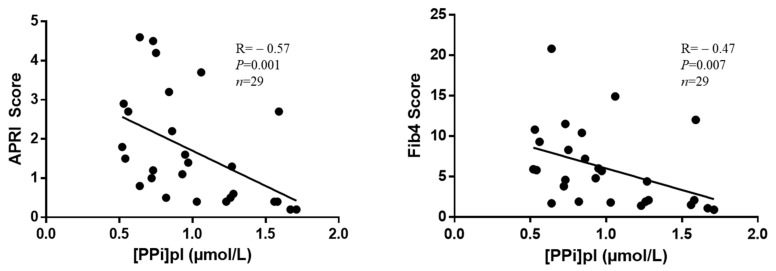
Correlations between [PPi]pl and the non-invasive indexes of liver fibrosis before LT. Legend: APRI and FIB-4 scores were calculated with variables measured at baseline. The correlations were assessed with the Spearman’s test.

**Figure 3 biomedicines-10-01496-f003:**
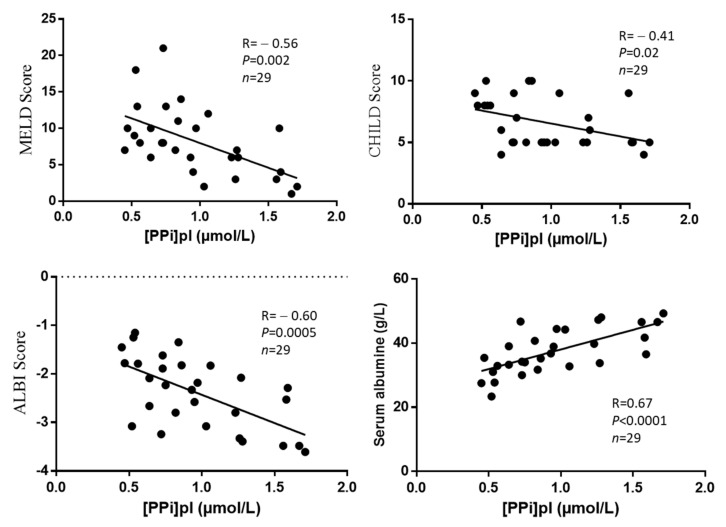
Correlations between [PPi]pl and the indexes of liver failure before LT. Legend: MELD, ALBI, and Pugh–Child scores were calculated with variables measured at baseline. The correlations were assessed with the Spearman test.

**Figure 4 biomedicines-10-01496-f004:**
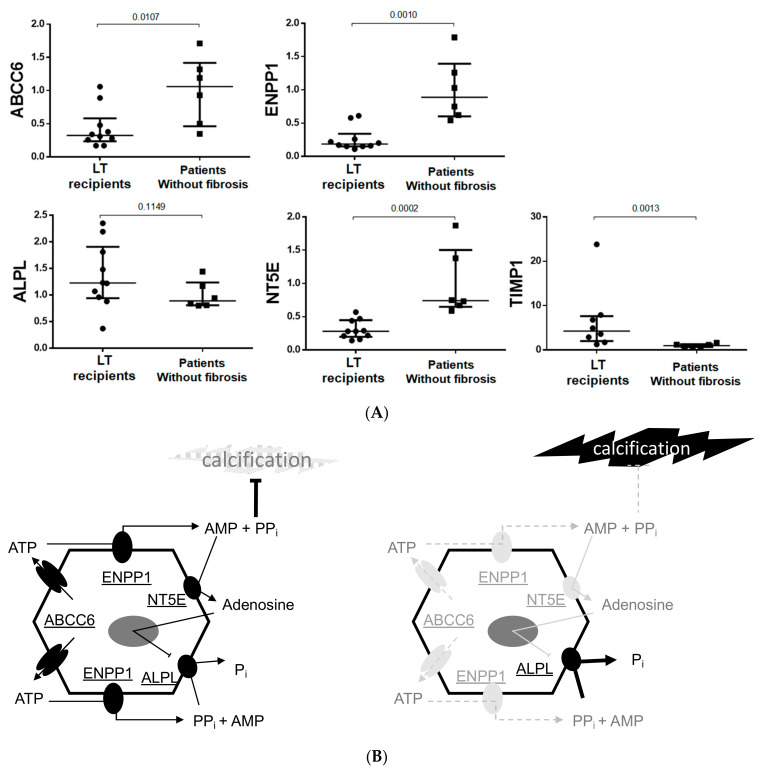
(**A**) mRNA levels of four genes regulating PPi homeostasis and of one gene related to fibrosis; (**B**) a schematic representation of PPi homeostasis in hepatocytes with respect to ectopic calcification. Legend 4(**A**): The figure shows the relative expression of genes directly involved in PPi homeostasis in the liver of LT candidates and controls. The gene expression values are normalized to RPLP0 mRNA levels. Results are expressed relative to the expression level in patients without fibrosis and statistically analyzed using the Mann–Whitney test. The results are shown as median ± interquartiles 25–75; Legend 4(**B**): The two diagrams are a representation of the relation between the PPi homeostasis in hepatocytes and arterial calcification. On the left panel (normal hepatocyte), PPi production and PPi hydrolysis are balanced, and the development of arterial calcifications is prevented. PPi production is initiated by the cellular release of ATP by ABCC6, which is then hydrolyzed by ENPP1 into PPi and AMP. PPi levels depend on NT5E, which generates adenosine from AMP. Adenosine normally inhibits the expression of ALPL, which encodes TNAP. In the right panel (hepatocyte in pathological liver), the decreased expression of ABCC6 and ENPP1 is consistent with the lower [PPi]pl we observed. Furthermore, sustain elevated plasma ALP activity enhances the PPi deficit and promotes arterial calcification.

**Figure 5 biomedicines-10-01496-f005:**
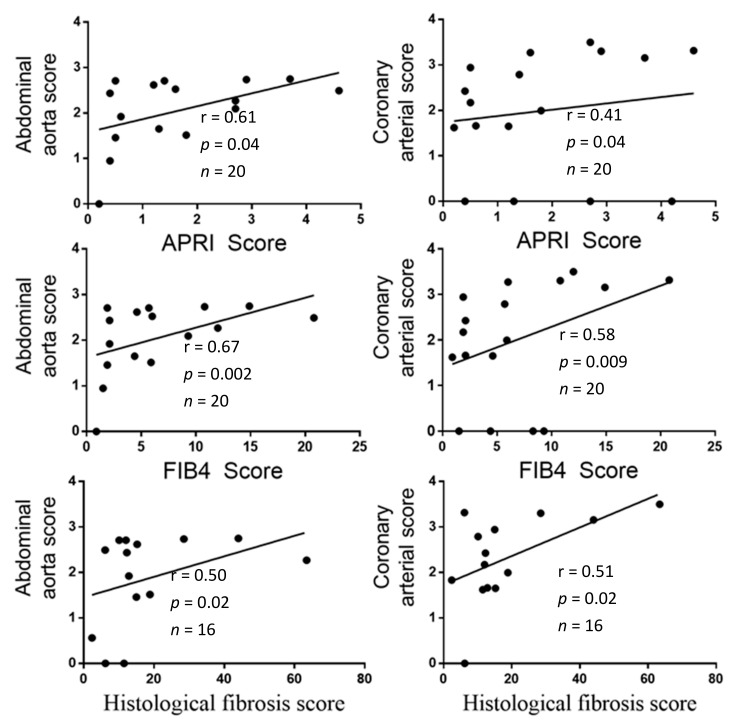
Correlations between arterial calcification scores and liver fibrosis indices in LTC. Legend: Liver fibrosis scores were calculated with variables measured at baseline (platelet count, AST, ALT) and the interstitial fibrosis index was measured on Masson trichrome-stained liver biopsies. The AC scores were measured on CT scans according to Agatston and log transformed. The length of the abdominal aorta was used to divide the abdominal aorta calcification score. The correlations were assessed with the Spearman test.

**Table 1 biomedicines-10-01496-t001:** Sequences of the primers. Legend: reverse and forward sequences of the primers used with with 2x SensiFAST SYBR HI-ROX mix.

Gene	Forward 5′→3′	Reverse 5′→3′
ABCC6	AAGGAACCACCATCAGGAGGAG	ACCAGCGACACAGAGAAGAGG
ENPP1	CCGTGGACAGAAATGACAGTTTC	ATGGACAGGACTAAGAGGAATTCTAAA
ALPL	TACAAGCACTCCCACTTCATCTG	GCTCGAAGAGACCCAATAGGTAGT
NT5E	GGGCGGAAGGTTCCTGTAG	GAGGAGCCATCCAGATAGACA
RPLP0	CAGATCCGCATGTCCCTTCG	AACACAAAGCCCACATTCCC

TaqMan gene expression assays were purchased from Thermo Fisher Scientific Inc.: RPLP0: Hs99999902_m1 and Timp1: Hs99999139_m1.

**Table 2 biomedicines-10-01496-t002:** Main characteristics of the liver transplant candidates.

Patients	Follow up of the Patients	Sex	Age (years)	BMI (kg/m²)	Pugh–Child	Fib-4	Liver Diseases	Hepatocellular Carcinoma
1	LTR dead before M3	F	51	24	5	1.10	relapsing hepatocellular carcinoma	yes
2	LTR available at M3	M	41	21	7	8.36	cryptogenetic	no
3	LTR available at M3	F	61	21	5	1.70	papillomatosis	no
4	LTC at M3	F	49	21	8	9.41	viral hepatitis C	yes
5	LTC at M3	M	53	26	7	4.46	alcohol	yes
6	LTC at M3	M	66	26	9	1.55	hemochromatosis	no
7	LTC at M3	M	58	27	6	2.10	viral hepatitis C and alcohol	yes
8	LTR available at M3	F	56	28	9	4.68	alcohol	no
9	LTC at M3	M	61	26	5	0.87	viral hepatitis C	yes
10	LTR available at M3	M	45	33	8	ND	alcohol	no
11	LTR available at M3	M	62	33	8	5.93	NASH	yes
12	LTR available at M3	M	66	21	5	1.95	viral hepatitis B	yes
13	LTC at M3	F	57	21	5	2.13	viral hepatitis C	yes
14	LTR available at M3	M	60	38	5	5.74	viral hepatitis C and alcohol	yes
15	LTC at M3	M	56	29	5	1.91	viral hepatitis C	yes
16	LTC at M3	M	59	30	9	15.0	alcohol	no
17	LTC at M3	M	63	38	5	6.02	alcohol and NASH	yes
18	LTR available at M3	M	63	30	10	10.8	alcohol and NASH	no
19	LTR available at M3	M	72	25	6	20.9	cryptogenetic	no
20	LTC at M3	F	57	28	5	12.1	viral hepatitis C and alcohol	yes
21	LTR available at M3	M	61	24	9		alcohol	yes
22	LTR available at M3	M	53	28	5	3.82	viral hepatitis C	yes
23	LTR dead before M3	M	51	27	5	1.41	viral hepatitis C and B	yes
24	LTC at M3	M	64	27	5	4.80	viral hepatitis C	yes
25	LTC at M3	F	57	31	10	10.5	NASH	no
26	LTC at M3	M	58	21	5	1.79	viral hepatitis C and alcohol	yes
27	Liver and kidney transplant recipient	M	58	21	5	11.5	viral hepatitis B, delta and alcohol	yes
28	LTR not available at M3	M	67	32	8	5.82	alcohol	yes
29	LTR available at M3	M	63	24	1	7.24	alcohol	no

Legend: LTC = liver transplant candidates; LTR = liver transplant recipients; BMI = body mass index; NASH = non-alcoholic steatohepatitis; M3 = month 3 (end of the study).

**Table 3 biomedicines-10-01496-t003:** Comparison between patients before and after LT.

Variables	LT Candidates (*n* = 29)	LT Recipients before LT (*n* = 12)	LT Recipients after LT (*n* = 12)	*p*
Interstitial fibrosis score (*n* = 16)	12.2 [7.2–17.9]	10.3 [5.2–16.1]	-	
FIB-4	4.7 [1.9–9.3]	5.8 [3.3–8.8]	-	
APRI	1.3 [0.5–2.7]	1.6 [1.0–3.2]	-	
MELD score	7 [3–12]	8 [5–13]	-	
ALBI score	−2.3 [−3.1–1.8]	−2.1 [−2.8–1.7]	-	
Pugh–Child score	6 [5–9]	8 [5–9]	-	
[PPi]pl (µmol/L)	0.86 [0.64–1.27]	0.68 [0.52–0.80]	1.40 [1.07–1.86]	0.0005
Fe PPi (%)	7.6 [4.0–15.0]	9.2 [5.7–18.8]	9.3 [5.7–17.2]	0.31
eGFR (mL/mn/1.73 m^2^)	94 [82–101]	92 [81–98]	73 [62–92]	0.002
Serum albumin (g/L)	37 [33–44]	35 [30–40]	40 [37–44]	0.009
C-reactiv protein (mg/L)	2 [1–8]	7 [2–12]	3 [1–8]	0.20
Plasma ALP activity (UI/L)	138 [76–245]	191 [87–322]	81 [65–156]	0.05
Prothrombin level (%)	76 [62–94]	75 [60–89]	100 [87–100]	0.01
Total bilirubin (µmol/L)	14 [10–30]	20 [10–44]	9 [8–13]	0.059
AST (UI/L)	46 [32–72]	54 [37–91]	23 [18–27]	0.008
ALT (UI/L)	34 [28–39]	37 [29–54]	26 [20–28]	0.04
GGT (UI/L)	55 [37–133]	98 [30–162]	24 [13–67]	0.11
Platelet count (×10^9^/L)	105 [71–216]	85 [72–142]	151 [124–233]	0.008

Legend: FIB-4 and APRI are non-invasive liver fibrosis indexes. The presence of liver fibrosis is indicated by a FIB-4 index > 3.25 or an APRI index > 1.5. MELD, ALBI, Pugh–Child Scores are liver failure indexes, which are predictive of mortality. Pugh–Child score between 7 and 9 predicts a 2-year survival of 60%. ALBI score shows an intermediate mortality risk. Meld score ≤ 9 is predictive of a three month mortality rate <1.9%. Three months after LT, immunosuppression consisted of a combination of mycophenolate mophetil with a reduced dose of tacrolimus and steroid (≤10 mg/d) in all patients. [PPi]pl = plasma level of inorganic pyrophosphate, FePPi = fractional excretion of inorganic pyrophosphate, ALP = alkaline phosphatase, eGFR = estimated glomerular filtration rate. Comparisons were performed with paired Mann–Whitney tests. LT candidates were not different from LT recipients before LT. *P* values are related to the comparisons between LT recipients before and 3 months post-surgery. Values are median ± interquartiles 25–75.

**Table 4 biomedicines-10-01496-t004:** Comparisons between LT candidates and biopsies from patients without liver fibrosis.

Variables	LT Recipients (*n* = 10)	Patients without Fibrosis (*n* = 6)	*p*
Pugh–Child score	8 [5–9]	-	
APRI	2.45 [0.83–3.63]	0.23 [0.20–0.30]	0.0004
FIB-4	5.78 [3.24–13.24]	0.43 [0.37–0.52]	0.0004
Plasma ALP activity (UI/L)	244 [110–364]	65 [58–83]	0.001
Total bilirubin (µmol/L)	35 [15–145]	6 [4–6]	0.003
Serum albumin (g/L)	29 [26–33]	46 [39–48]	0.002

Legend: FIB-4 and APRI are liver fibrosis indexes, Pugh–Child Scores is a liver failure index, ALP = alkaline phosphatase. Comparisons were performed with Mann–Whitney tests. Values are median ± interquartiles 25–75.

## Data Availability

Data is contained within the article.

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
