# Peer review of "Arterial Calcifications in Patients with Liver Cirrhosis Are Linked to Hepatic Deficiency of Pyrophosphate Production Restored by Liver Transplantation"

_biomedicines, 2022, doi:10.3390/biomedicines10071496_

Round 1

Reviewer 1 Report

This is a very interesting article on the processes of arterial calcification in patients with alcoholic cirrhosis that lead to liver transplantation. However, the analyzed cohort is insufficient to obtain conclusive results and clear conclusions.

Several aspects should be clarified in the manuscript:

  1. It is not specified what type of immunosuppressive agent is used after transplantation. (tacrolimus, everolimus or cycloporine…). During the first months to avoid rejection, they are usually used in high doses and can influence the mineral metabolism of the receiving organism. This aspect should be considered by the authors.
  2. The abstract with its objective should be more explanatory, including the populations analyzed in the study.
  3. Abbreviations must be described the first time they appear in the text.
  4. The use of connectors in the writing would speed up the reading of the manuscript.
  5. Patients section: The legislative part could go together. This section should explain in more detail why the control group are obese people.
  6. All the variables and scales analyzed in the study must be referenced and the units of measurement should be indicated. The statistics section is very poor. Many scales need to be developed in the manuscript.
  7. The authors could explain the use of the different biological matrices used in the schedule, as well as the techniques used.
  8. Why don't the authors use Fibroscan to grade liver fibrosis?
  9. Genetic expression data should be provided, as well as the primers used in the analyses.
  10. The clinical, biochemical description of the patients, it may be advisable to show them in a table.
  11. Remove Liver function test, it is more appropriate to talk about sociodemographic, clinical and biochemical parameters.
  12. Table 1. Scale ranges should be shown in figure captions. In the case of post-transplantation, the types of immunosuppressants used should be indicated.
  13. It is recommended to indicate in the results when pre-transplant data is shown, post-transplant or control.
  1. In FIG. 4, TIMP1 is not shown.

Author Response

COMMENT 1

It is not specified what type of immunosuppressive agent is used after transplantation. (tacrolimus, everolimus or cycloporine…). During the first months to avoid rejection, they are usually used in high doses and can influence the mineral metabolism of the receiving organism. This aspect should be considered by the authors.

We thank the reviewer for his/her important comment.

The usual immunosuppression protocol in our centre consisted of 1 injection in the operating room during the implantation of the liver graft of 4.5 mg/kg of IV methylprednisolone. This dose was repeated once on arrival in the intensive care unit on leaving the operating theatre.

Then, from day 1 post-op, oral corticosteroid therapy of 20 mg/day of prednisolone was started. In the absence of complications such as rejection, this was then regularly reduced by 2 mg in 2 mg, every week, from 15/21 days post-op.

All patients included in the study received standard immunosuppression consisting, in our centre, of a combination of mycophenolate mophetil with a reduced dose of tacrolimus. This strategy has been proposed in the liver transplantation literature for the past 15 years as being associated with fewer long-term renal complications compared to the use of full-dose tacrolimus. The use of reduced doses of an anti-calcineurin molecule is a concept widely applied in liver transplantation.

Except for specific complications (bone marrow toxicity), patients were kept on this combination (low dose tacrolimus + mycophenolate mophetyl) for life.

In patients who had acute or chronic renal failure before liver transplantation or renal failure immediately after liver transplantation, a delayed introduction of tacrolimus was carried out at D7. Immediate immunosuppression was then ensured by induction with IV basiliximab administered at D1 and D5.

None of the included patients received everolimus or cyclosporine.

COMMENT 2

The abstract with its objective should be more explanatory, including the populations analyzed in the study.

We provided more precision in the paragraph “objectives”, as follows:

To that aim, we compared [PPi]pl before liver transplantation (LT) and 3 months after LT. We also assessed the expression of 4 key regulators of PPi in liver tissues and established correlations between arterial calcifications (AC), indices of liver fibrosis and liver failure in these patients.

We detailed the populations analyzed in the study in the paragraph “methods”, as follows:

LT candidates with various liver diseases were included. AC scores were assessed in coronary arteries, abdominal aorta and aortic valves. Liver fibrosis was evaluated on liver biopsies and from non-invasive tests (FIB-4 and APRI scores). Liver functions were assessed by measuring serum albumin, ALBI, MELD and Child-Pugh scores. An enzymatic assay was used to dose [PPi]pl. A group of patients without liver alterations from a previous cohort provided a control group. Gene expression assays were performed with mRNA extracted from liver biopsies and compared between LT recipients and the control individuals.

COMMENT 3

Abbreviations must be described the first time they appear in the text.

We paid attention to this comment and proofed the revised manuscript.

COMMENT 4

The use of connectors in the writing would speed up the reading of the manuscript.

We did our best to improve the reading of the revised manuscript.

COMMENT 5

In patient section, the legislative part could go together.

This is not possible, because there are two different legislative parts (study patients and obese cohort).

COMMENT 6

This section should explain in more detail why the control group are obese people.

Access to liver biopsies from healthy subjects is extremely limited. We modified the paragraph in patient section as follows:

To constitute a group of liver biopsies without liver alterations, we took advantage of our cohort of morbidly obese patients. These patients were recruited through the Department of Digestive Surgery and Liver Transplantation (Archet 2, University Hospital, Nice, France) where they underwent bariatric surgery for their morbid obesity. All subjects gave informed written consent in accordance with French legislation on Ethics and Human Research referred to as Huriet-Serusclat’s law. The ‘‘Comité Consultatif de Protection des Personnes dans la Recherche Biomédicale de Nice” approved the study (07/04/2003, N° 03.017). Surgical liver biopsies were obtained during surgery and no ischemic preconditioning was performed. Hepatic histopathological analysis was performed according to the scoring system of Kleiner et al. . Liver samples were selected from patients without hepatitis B or hepatitis C infection, excessive alcohol consumption (>20g/d) or another cause of chronic liver disease as previously described. Liver samples did not display any hepatic steatosis, inflammation or fibrosis.

COMMENT 7

All the variables and scales analyzed in the study must be referenced and the units of measurement should be indicated. The statistics section is very poor. Many scales need to be developed in the manuscript.

Clinical scoring systems, based on simple clinical or laboratory indices, are used to identify liver fibrosis (Fib4 and APRI) or the severity of the liver failure according to the mortality rate (Albi, Meld, Child-Pugh). The threshold values of Fib4 and Apri for the prediction of liver fibrosis are established with receiver operating characteristic (ROC) curves in patients with advanced histological liver fibrosis.

We indicated a reference for every scoring system in the revised manuscript (paragraph calculation of scores) and commented the results from Table 1 in the legend as follows:

FIB-4 and APRI are non-invasive liver fibrosis indexes. The presence of liver fibrosis is indicated by a FIB-4 index > 3.25 or an APRI index > 1.5.

MELD, ALBI, Pugh-Child Scores are liver failure indexes, which are predictive of mortality. Pugh-Child score between 7 and 9 predicts a 2-year survival of 60%. ALBI score indicates an intermediate mortality risk. Meld score ≤ 9 is predictive of a mortality rate <1.9% at three month.

We added these informations to the legend of Table 1.

Regarding statistics, only basic approaches were possible in this pilot study (description, correlation only) due to the limited number of patients (the limited number of patients does not support the search for confounding variables).

COMMENT 8

The authors could explain the use of the different biological matrices used in the schedule, as well as the techniques used.

Please, fin below more details regarding the matrices and the techniques:

Liver tissue was used to assess the interstitial fibrosis with histochemistry and to assess the expression of genes with RT-qPCR.

Plasma was used to measure pyrophosphate levels with an enzymatic assay and ALP activity was assessed with an absorbance method.

Serum was used to measure the creatinine levels, which is necessary to calculate the renal function with the CKD-Epi formula.

Clinical and biological indices were used to compute the non-invasive scores of liver fibrosis (Fib-4, Apri) and the scores of the severity of liver failure (Meld, Albi, Pugh-Child).

Computer tomograms were used to measure the arterial calcifications with an appropriate software (4DM).

COMMENT 9

Why don't the authors use Fibroscan to grade liver fibrosis?

FibroScan was not performed in this study. We did not consider it useful when designing the study. Indeed, all LTR had explant histology. The diagnosis of cirrhosis was made on the basis of clinical-biological-imagery evidence and sometimes by liver biopsy. In addition, it is very difficult to perform FibroScan in patients with ascites. This would have made it difficult to interpret the FibroScan variation at the intra-individual level.

COMMENT 10

Genetic expression data should be provided, as well as the primers used in the analyses.

The 2–ΔΔCt for each gene are shown on Fig.4A. Please, find below a table with the ΔCt for each gene:

ΔCt

ΔCt

ΔCt

ΔCt

ΔCt

ABCC6

ENPP1

ALPL

NT5E

TIMP1

patients without fibrosis

LTR

patients without fibrosis

LTR

patients without fibrosis

LTR

patients without fibrosis

LTR

patients without fibrosis

LTR

25% Percentile

2,10

3,42

1,07

3,19

3,64

3,01

1,37

3,10

1,63

-0,85

Median

2,50

4,20

1,71

3,97

4,11

3,65

2,38

3,80

2,06

0,29

75% Percentile

3,73

4,69

2,26

4,28

4,25

4,03

2,57

4,30

2,56

1,31

Please, find below the sequences of the primers used in the analysis:

The sequence of the primers used with 2x SensiFAST SYBR HI-ROX mix (Bioline) were as follows: ABCC6 forward 5’-AAGGAACCACCATCAGGAGGAG-3’, reverse 5’-ACCAGCGACACAGAGAAGAGG-3’; ENPP1 forward 5’-CCGTGGACAGAAATGACAGTTTC-3’, reverse 5’-ATGGACAGGACTAAGAGGAATTCTAAA-3’; ALPL forward 5’-TACAAGCACTCCCACTTCATCTG-3’, reverse 5’-GCTCGAAGAGACCCAATAGGTAGT-3’; NT5E forward 5’-GGGCGGAAGGTTCCTGTAG-3’, reverse 5’-GAGGAGCCATCCAGATAGACA-3’; RPLP0 forward 5’-CAGATCCGCATGTCCCTTCG-3’, reverse 5’-AACACAAAGCCCACATTCCC-3’.

TaqMan gene expression assays were purchased from Thermo Fisher Scientific Inc: RPLP0: Hs99999902_m1 and Timp1: Hs99999139_m1

We rephrased the paragraph regarding RTPCR in the revised manuscript as follows:

Gene expression was normalized to the RPLP0 (Ribosomal Phosphoprotein Large P0) housekeeping gene and calculated based on the comparative cycle threshold Ct method (2–ΔΔCt). The results were expressed relative to mRNA levels in “Patients without fibrosis”. Results are expressed as median ± interquartiles 25-75 and statistically analyzed using the Mann-Whitney test.

We modified the legend from Fig 4A as follows:

The figure shows the relative expression of genes directly involved in PPi homeostatis in the liver of LT candidates and controls. The gene expression values are normalized to RPLP0 mRNA levels. Results are expressed relative to the expression level in patients without fibrosis and statistically analyzed using the Mann–Whitney test. The results are shown as median ± interquartiles 25-75.

COMMENT 11

The clinical, biochemical description of the patients, it may be advisable to show them in a table.  

Please, find below a table with individual data from the patients.

patient

Follow up of the patients

sex

Age

BMI

Child

Fib4 

liver disease

Hepatocellular carcinoma

1

LTR dead before M3

F

51

24

5

1.10

relapsing hepatocellular carcinoma

yes

2

LTR available at M3

M

41

21

7

8.36

cryptogenetic

no

3

LTR available at M3

F

61

21

5

1.70

Papillomatosis

no

4

LTC at the end of the study

F

49

21

8

9.41

viral hepatitis C

yes

5

LTC at the end of the study

M

53

26

7

4.46

alcohol

yes

6

LTC at the end of the study

M

66

26

9

1.55

Hemochromatosis

no

7

LTC at the end of the study

M

58

27

6

2.10

viral hepatitis C and  alcohol

yes

8

LTR available at M3

F

56

28

9

4.68

alcohol

no

9

LTC at the end of the study

M

61

26

5

0.87

viral hepatitis C

yes

10

LTR available at M3

M

45

33

8

ND

alcohol

no

11

LTR available at M3

M

62

33

8

5.93

NASH

yes

12

LTR available at M3

M

66

21

5

1.95

viral hepatitis B

yes

13

LTC at the end of the study

F

57

21

5

2.13

viral hepatitis C

yes

14

LTR available at M3

M

60

38

5

5.74

viral hepatitis C and  alcohol

yes

15

LTC at the end of the study

M

56

29

5

1.91

viral hepatitis C

yes

16

LTC at the end of the study

M

59

30

 9

15.0

alcohol

no

17

LTC at the end of the study

M

63

38

5

6.02

alcohol and NASH

yes

18

LTR available at M3

M

63

30

10

10.8

alcohol and NASH

no

19

LTR available at M3

M

72

25

6

20.9

cryptogenetic

no

20

LTC at the end of the study

F

57

28

5

12.1

viral hepatitis C and  alcohol

yes

21

LTR available at M3

M

61

24

9

alcohol

yes

22

LTR available at M3

M

53

28

5

3.82

viral hepatitis C

yes

23

LTR dead before M3

M

51

27

5

1.41

viral hepatitis C and B

yes

24

LTC at the end of the study

M

64

27

5

4.80

viral hepatitis C

yes

25

LTC at the end of the study

F

57

31

10

10.5

NASH

no

26

LTC at the end of the study

M

58

21

5

1.79

viral hepatitis C and  alcohol

yes

27

Liver and kidney transplant recipient

M

58

21

5

11.5

viral hepatitis B, delta and alcohol

yes

28

LTR not available at M3

M

67

32

 8

5.82

alcohol

yes

29

LTR available at M3

M

63

24

1

7.24

alcohol

no

COMMENT 12

Remove Liver function test, it is more appropriate to talk about sociodemographic, clinical and biochemical parameters.

we paid attention to this comment and line 248 we rephrased “sociodemographic, clinical and biochemical parameters”

COMMENT 13

Table 1. Scale ranges should be shown in figure captions.

We added these informations in the legend:

Legend: FIB-4 and APRI are non-invasive liver fibrosis indexes. The presence of liver fibrosis is indicated by a FIB-4 index > 3.25 or an APRI index > 1.5. MELD, ALBI, Pugh-Child Scores are liver failure indexes, which are predictive of mortality. Pugh-Child score between 7 and 9 predicts a 2-year survival of 60%. ALBI score indicates an intermediate mortality risk. Meld score ≤ 9 is predictive of a mortality rate <1.9% at three month.

COMMENT 14

In the case of post-transplantation, the types of immunosuppressants used should be indicated.

We added the following sentence in the legend of Table 1: “Three months after LT, immunosuppression consisted of a combination of mycophenolate mophetil with a reduced dose of tacrolimus and steroid (≤10 mg/d)”.

COMMENT 15

It is recommended to indicate in the results when pre-transplant data is shown, post-transplant or control.

We paid attention to this comment :

Lines 268, 272 , 276, 279:  “before LT” was added.

Line 288: “downregulated in LT recipients” was changed in “downregulated in the diseased liver of LT recipients”

Lines 314, 321, 330 “in LTC” was added

COMMENT 16

In FIG. 4, TIMP1 is not shown.  

We increased the size of figure 4A because the letters of labels were too small to read and the expression of TIMP1 could be overwieved.

Reviewer 2 Report

The manuscript is interesting, however there are some questions below.

  1. In Introduction section, “Liver transplantation (LT) is the treatment of liver cirrhosis with end stage liver failure”?
  2. “Exclusion criteria included the presence of hepatitis B virus (HBV) or hepatitis C virus (HCV) infection, excessive…”?
  3. In Specific dosages section, please show the primer sequences or TaqMan Probes? Which did you use the enzymes?
  4. What are the cause of diseases in the study patients? Authors should show them clearly.
  5. In Figures 4 and 6, letters of labels are too small to read.
  6. In Table 1, how the renal function??

Author Response

COMMENT 1

In Introduction section, “Liver transplantation (LT) is the treatment of liver cirrhosis with end stage liver failure”?

We thank the reviewer for this helpful comment. LT is not solely recommended in end stage liver failure due to chronic liver diseases with cirrhosis. Actually, LT may be performed for the purpose of treating hepatocellular carcinoma. Indeed, in our small cohort, many patients had hepatocellular carcinoma in addition to cirrhosis. For more clarity, please find a table with individual data regarding the patients. One patients without cirrhosis and with papillomatosis was also present in our cohort.

In the revised manuscript, we modified the introduction as follows: “Liver cirrhosis with end stage liver failure and/or hepatocellular carcinoma may be treated with liver transplantation (LT).”

patient

Follow up of the patients

sex

Age

BMI

Child

Fib4 

liver disease

Hepatocellular carcinoma

1

LTR dead before M3

F

51

24

5

1.10

relapsing hepatocellular carcinoma

yes

2

LTR available at M3

M

41

21

7

8.36

cryptogenetic

no

3

LTR available at M3

F

61

21

5

1.70

Papillomatosis

no

4

LTC at the end of the study

F

49

21

8

9.41

viral hepatitis C

yes

5

LTC at the end of the study

M

53

26

7

4.46

alcohol

yes

6

LTC at the end of the study

M

66

26

9

1.55

Hemochromatosis

no

7

LTC at the end of the study

M

58

27

6

2.10

viral hepatitis C and  alcohol

yes

8

LTR available at M3

F

56

28

9

4.68

alcohol

no

9

LTC at the end of the study

M

61

26

5

0.87

viral hepatitis C

yes

10

LTR available at M3

M

45

33

8

ND

alcohol

no

11

LTR available at M3

M

62

33

8

5.93

NASH

yes

12

LTR available at M3

M

66

21

5

1.95

viral hepatitis B

yes

13

LTC at the end of the study

F

57

21

5

2.13

viral hepatitis C

yes

14

LTR available at M3

M

60

38

5

5.74

viral hepatitis C and  alcohol

yes

15

LTC at the end of the study

M

56

29

5

1.91

viral hepatitis C

yes

16

LTC at the end of the study

M

59

30

 9

15.0

alcohol

no

17

LTC at the end of the study

M

63

38

5

6.02

alcohol and NASH

yes

18

LTR available at M3

M

63

30

10

10.8

alcohol and NASH

no

19

LTR available at M3

M

72

25

6

20.9

cryptogenetic

no

20

LTC at the end of the study

F

57

28

5

12.1

viral hepatitis C and  alcohol

yes

21

LTR available at M3

M

61

24

9

alcohol

yes

22

LTR available at M3

M

53

28

5

3.82

viral hepatitis C

yes

23

LTR dead before M3

M

51

27

5

1.41

viral hepatitis C and B

yes

24

LTC at the end of the study

M

64

27

5

4.80

viral hepatitis C

yes

25

LTC at the end of the study

F

57

31

10

10.5

NASH

no

26

LTC at the end of the study

M

58

21

5

1.79

viral hepatitis C and  alcohol

yes

27

Liver and kidney transplant recipient

M

58

21

5

11.5

viral hepatitis B, delta and alcohol

yes

28

LTR not available at M3

M

67

32

 8

5.82

alcohol

yes

29

LTR available at M3

M

63

24

1

7.24

alcohol

no

COMMENT 2

“Exclusion criteria included the presence of hepatitis B virus (HBV) or hepatitis C virus (HCV) infection, excessive…”?

This paragraph relates to the control group of patients without liver disease, lines 106-108 of the manuscript, and not to the LT candidates (13 had viral hepatitis). We rephrased the paragraph “patients” for more clarity.

COMMENT 3

In Specific dosages section, please show the primer sequences or TaqMan Probes? Which did you use the enzymes?  

Please, find below the sequences of the primers used in the analysis:

The sequence of the primers used with 2x SensiFAST SYBR HI-ROX mix (Bioline) were as follows: ABCC6 forward 5’-AAGGAACCACCATCAGGAGGAG-3’, reverse 5’-ACCAGCGACACAGAGAAGAGG-3’; ENPP1 forward 5’-CCGTGGACAGAAATGACAGTTTC-3’, reverse 5’-ATGGACAGGACTAAGAGGAATTCTAAA-3’; ALPL forward 5’-TACAAGCACTCCCACTTCATCTG-3’, reverse 5’-GCTCGAAGAGACCCAATAGGTAGT-3’; NT5E forward 5’-GGGCGGAAGGTTCCTGTAG-3’, reverse 5’-GAGGAGCCATCCAGATAGACA-3’; RPLP0 forward 5’-CAGATCCGCATGTCCCTTCG-3’, reverse 5’-AACACAAAGCCCACATTCCC-3’.

TaqMan gene expression assays were purchased from Thermo Fisher Scientific Inc: RPLP0: Hs99999902_m1 and Timp1: Hs99999139_m1

The probes have been used following the manufacturer’s protocol

COMMENT 4

What are the cause of diseases in the study patients? Authors should show them clearly.

Chronic liver diseases of the study patients are listed lines 237-243 of the manuscript. For more clarity, we added the causes of the liver diseases of each patient in the aforementioned table.

COMMENT 6

In Figures 4 and 6, letters of labels are too small to read.

Fig 4A and 5 were changed to be more readable.

COMMENT 7

In Table 1, how the renal function?? The renal function is normal, as indicated by the estimated glomerular filtration rate (eGFR) in table 1

Reviewer 3 Report

The present study is an interesting addition to current understanding of the coupling between liver fibrosis,  PPi homeostasis, and arterial calcification.  Generally, despite the limited case number and simple statistical correlations alone, most of the key phrases and concepts have been clearly addressed in Introduction, the assays clearly described, in-depth genetic expression profiles included, and longitudinal, pretreatment-posttreatment  analyses on the invaluable dataset over time well implemented.

Comments:
Introduction section: lines 59, 60, 61:
-one of the domain thrusts-- connection between serum or plasma alkaline phosphatase activity and liver cirrhosis does not sound familiar to clinical hepatologists
-citation 9 is old and weak; cite a latest reference
-may be appropriate to address more discusssions based on more references relevant to this connection/gap than the present in Discussion section      

Author Response

COMMENTS
Introduction section: lines 59, 60, 61:
-one of the domain thrusts-- connection between serum or plasma alkaline phosphatase activity and liver cirrhosis does not sound familiar to clinical hepatologists
-citation 9 is old and weak; cite a latest reference
-may be appropriate to address more discussions based on more references relevant to this connection/gap than the present in Discussion section      

We thank the reviewer for this important comment. We provided more recent references regarding the plasma alkaline phosphatase activity and discussed the connection between ALP activity and liver cirrhosis accordingly.

In the introduction, we added the following paragraph :

“Plasma alkaline phosphatase (ALP) activity results from a mixture of bone and liver ALP, which are produced by the same ALPL gene encoding tissue non-specific ALP (TNAP). In the absence of bone disease and vitamin D deficiency, a mild increase of plasma ALP activity is observed in many liver diseases, and a marked elevation of plasma ALP activity is the signpost of the primary cholestatic diseases.”

We removed ref 9 and added ref Poupon (hepatology 2015) and ref Corpechot (NEJM 2018), which are ref 9 and 10 in the revised manuscript.

It would be interesting to compare patients with primary cholestatic diseases (primary sclerosing cholangitis and primary biliary cholangitis) and patients with other chronic liver diseases regarding plasma PPI levels and arterial calcifications.

Round 2

Reviewer 1 Report

The article is correct in its present form.

Author Response

We thank the reviewer for his/her comment.

Reviewer 2 Report

Please include the patients' list and primers list.

Please make changes to Journal format. ex) take out legends under the tables.

Author Response

We thank the reviewer for his/her recommandations.

We added the primer sequences and the table with the main individual characteristics of the patients (Table 1 of the revised manuscript). We re-numbered the other tables accordingly.

We went through the author guidelines and did not find the recommandation not put the legends under the tables. So we did not modify the place of the legend. However, we put the titles before the figures and the legends below.

We hope this is convenient.